# Natural History of Ulcerative Colitis with Coexistent Colonic Diverticulosis

**DOI:** 10.3390/jcm10061192

**Published:** 2021-03-12

**Authors:** Irene Marafini, Silvia Salvatori, Irene Rocchetti, Norma Alfieri, Patrizio Scarozza, Emma Calabrese, Livia Biancone, Giovanni Monteleone

**Affiliations:** 1Gastroenterology Unit, Department of Systems Medicine, University of Rome Tor Vergata, 00133 Rome, Italy; irene.marafini@gmail.com (I.M.); silviasalvatori23@gmail.com (S.S.); norma.alfieri@outlook.com (N.A.); scarozzapatrizio@gmail.com (P.S.); emma.calabrese@uniroma2.it (E.C.); biancone@med.uniroma2.it (L.B.); 2Statistical Office, Superior Council of Judiciary, 00185 Rome, Italy; irene.rocchetti@gmail.com

**Keywords:** inflammatory bowel disease, diverticular disease, diverticula, steroid-dependence

## Abstract

Ulcerative colitis (UC) and colonic diverticulosis can co-exist in some patients. However, the natural history of UC associated with colonic diverticulosis is not well known. We here compared the disease characteristics and outcome of UC patients with and without concomitant colonic diverticulosis. Medical records of 347 UC patients were included in an observational, retrospective, nested-matched case-control study. Cases were 92 patients with UC and concomitant colonic diverticulosis, while controls were 255 UC patients without concomitant colonic diverticulosis. A propensity score matching (PSM) was used to homogenate cases (n = 92) and controls (n = 153) for age. UC patients with concomitant colonic diverticulosis were less likely to have an extensive disease (25/92, 27.1%) and to experience steroid dependence (8/92, 8.6%) compared to patients without concomitant colonic diverticulosis (70/153, 45.7% and 48/153, 31.3%, respectively; *p* < 0.001). The use of immunosuppressants (9/92, 9.7% vs. 37/153, 24.1%; *p* = 0.007) or biologics (3/92, 3.2% vs. 26/153, 16.9%, *p* < 0.001) was significantly lower in UC patients with concomitant diverticulosis compared to the control group. On multivariate analysis, steroid dependence and extensive colitis were significantly less frequent in UC patients with concomitant colonic diverticulosis compared to UC patients without diverticula. UC patients with coexisting colonic diverticulosis are less likely to have an extensive disease and to be steroid-dependent.

## 1. Introduction

Ulcerative colitis (UC) and colonic diverticulosis are distinct clinical conditions that affect the large intestine. UC is a chronic immune-mediated disorder of the gastrointestinal tract of unknown aetiology, in which inflammation starting from the rectum can extend proximally and involve the whole colon [1]. UC has a bimodal pattern of incidence, with the main onset peak between 15 and 30 years of age and a second, smaller peak between 50 and 70 years of age [1]. UC patients experience episodes of relapse during their disease course and in some cases can require steroids, conventional immunosuppressants, and biologic therapies [2]. Colonic diverticulosis is a multifactorial acquired clinical condition, very common in developed countries and characterized by the presence of small out-pouchings from the colonic lumen caused by the mucosal herniation through the colonic wall at sites of vascular perforation [3]. In the majority of cases, colonic diverticulosis remains asymptomatic, while 10–20% of patients become clinically symptomatic and few of them develop complications such as acute inflammation, bleeding, and perforation [3].

Conflicting results have been published about the prevalence and the clinical relevance of colonic diverticulosis in UC patients. Although most authors reported a lower prevalence of colonic diverticulosis in UC patients as compared to the general population [4,5,6], some studies documented an increased frequency of coexistence of these two conditions [7]. It was also shown that UC patients with colonic diverticulosis have a higher mean age at diagnosis of UC compared to patients without colonic diverticulosis [8]. In 2018, Nascimbeni and co-workers performed an observational, case-control study aimed at assessing prevalence, features, and risk factors of colonic diverticula in patients with UC compared to a control health population undergoing screening colonoscopy [9]. The authors showed that colonic diverticulosis was less frequent in UC patients compared to controls of same age and gender. As expected, advancing age was a risk factor for the development of diverticula in both groups. Among UC patients, the presence of diverticula was associated with a short duration and a late onset of UC. Moreover, UC patients with diverticula had a significantly lower frequency of annual disease flares per year and a trend toward more frequent extension of UC to the left colon [9]. In contrast, Kinnucan and colleagues documented a slightly higher prevalence of colonic diverticulosis in UC patients compared to control subjects and no difference between the two groups in terms of disease duration and extent [7]. The reasons for such discrepancies remain unknown even though differences in the selection of cases and controls, the type of statistical analysis, and the duration of follow-up could have well contributed. 

In this study, we compared the characteristics and course of UC patients with and without concomitant colonic diverticulosis. 

## 2. Materials and Methods

### 2.1. Study Population and Data Source

We performed an observational, retrospective, single-centre, nested-matched case-control study. All patients gave their written consent to scientific and anonymous use of their clinical data at the time of enrolment into study database. Since advancing age is associated with colonic diverticulosis, we enrolled patients with age ≥ 45 years, who had a confirmed diagnosis of UC according to the standard criteria [10] and were in a regular follow-up at a single university hospital (University of Rome “Tor Vergata”). Colonic diverticulosis was diagnosed with colonoscopy and its prevalence was determined by the presence or the absence of colonic diverticula independently of their number or pathologic features. Exclusion criteria in both groups of patients included a diagnosis of Crohn’s disease or microscopic colitis. Patients with missing data regarding duration and extension of UC were excluded. Patients with colonic diverticula located in the right colon only were also excluded from the analysis. For each patient, the following variables were collected: gender, age, smoking habit, disease’s duration, history of appendectomy, UC extent, steroid-dependence/refractoriness, necessity of immunosuppressive/biologic therapy, number of relapses per year, hospitalizations, surgeries, presence of concomitant colonic diverticulosis, and duration of follow-up. For patients undergoing colonic surgery for complicated diverticular disease, the number of flares was considered before and after colonic resections. UC patients with concomitant colonic diverticulosis were assigned to the case group, while UC patients without colonic diverticulosis were included in the control group. We performed a nested case-control analysis of this sample. Since old age is one of the most relevant risk factors for the development of colonic diverticulosis, patients in case and control groups were matched according to their age. 

### 2.2. Statistical Analysis 

A propensity score matching (PSM) was used to homogenate cases and controls according to their age. Then, the two groups obtained from the PSM were compared by using the *t* test and the χ-square test. For bimodal variables, the test of the difference between the proportions was also applied to understand whether the frequency of the disease characteristics among the cases was greater or lower with respect to those of the controls, and the statistical significance of the eventual inequality was computed. *p* values ≤ 0.05 were considered to assess the statistical significance. A multivariate analysis was applied to estimate the probability for units having data characteristics to be a case rather than a control.

## 3. Results

### 3.1. Case-Control Propensity Score Matching

At the time of the study, our inflammatory bowel disease (IBD) database comprised 1976 UC patients. After applying eligibility and exclusion criteria, the UC group consisted of 347 patients. There were 217 males and 130 females. In total, 1629 patients were not included for the following reasons: age < 45 years in 1602 patients, UC diagnosis within 1 year preceding study in 24 patients, colonic diverticulosis located in the right colon only in three patients. Cases were 92 patients with concomitant colonic diverticulosis while control group included 255 patients without colonic diverticulosis. At the time of enrolment, the median age in the cases was 71 years (range 48–92 years) and in the controls was 63 years (range 33–89 years). Upon the application of the PSM, the 92 cases were matched with 153 controls in 1:3 ratio, while the remaining 102 controls were excluded from the subsequent analyses. 

### 3.2. Characteristics of Colonic Diverticulosis in UC Patients

Characteristics of colonic diverticulosis in UC patients are shown in Table 1. Colonic diverticulosis was diagnosed before the diagnosis of UC in 14/92 patients (15.2%), after UC in 57/92 patients (61.9%), and concomitantly to UC in 21/92 patients (22.8%). In most patients (n = 70; 76.0%), diverticula were localised in the left colon, while in 22/92 patients, diverticula were localized in the whole colon (23.9%). Episodes of acute diverticulitis were documented in 11/92 (11.6%) patients, and 4/92 patients (4.2%) underwent colorectal surgery for complicated diverticular disease. Surgeries consisted in left hemicolectomy in three cases and one segmental colonic resection.

### 3.3. Ulcerative Colitis Patients with Concomitant Colonic Diverticulosis Have Prevalently a Mild Disease Course

After case-control PSM, 92 cases and 153 controls were compared for the disease characteristics (Table 2). The two groups were homogenous for age and gender. Moreover, there was no significant difference between the two groups in the number of follow-up years (7 and 5 median years; *p* = 0.9). The number of current (10/92, 10.8%; 11/153, 7.1%) and former smokers (32/92, 34.7%; 46/153, 30.6%) did not differ between the two groups (*p* = 0.11). There was a higher but not statistically different number of appendectomies in the case group (10/92, 10.8%) compared to the controls (6/153, 3.9%; *p* = 0.056). Mean disease duration was 19 years in cases and 16 years in controls (*p* = 0.008). Among the cases, 19/92 (20.6%) patients had proctitis, 48/92 (52.1%) patients had left-sided colitis, and 25/92 (27.1%) patients had extensive colitis, while in the control group, 14/153 (9.1%) patients had proctitis, 69/153 (45%) patients had left colitis, and 70/153 (45.7%) patients had extended colitis. Therefore, case and control groups differ significantly for the number of patients with proctitis (*p* < 0.001) and extensive colitis (*p* < 0.001), while no differences emerged from the comparison of patients with left-sided colitis (*p* = 0.068; Table 2). Steroid dependence was significantly less frequent in the cases (8/92; 8.6%) compared to the controls (48/153; 31.3%, *p* < 0.001), while there was no difference in terms of steroid refractoriness (1/92, 1% vs. 7/153, 4.5%; *p* = 0.09). During their disease course, UC patients with concomitant colonic diverticulosis had a statistically lower need of immunosuppressants compared to controls (9/92 (9.7%) vs. 37/153 (24.1%); (*p* = 0.007)). Moreover, the need of biologics was significantly reduced in the cases as compared to controls (3/92 (3.2%) vs. 26/153 (16.9%); (*p* < 0.001)). The number of patients with more than one relapse per year was significantly lower in the cases compared to controls (6/92 (6.5%) vs. 16/153 (10.4%); (*p* = 0.013)). 

Moreover, there were no differences between the two groups for the number of surgeries (13/92 (14.1%) vs. 13/153 (8.4%); (*p* = 0.068)), hospitalizations (28/92 (30.4%) vs. 40/153 (26.1%); *p* = 0.44)) and concomitant extra-intestinal manifestations (12/92 (13%) vs. 32/153 (20.9%); (*p* = 0.17)). On multivariate analysis, steroid dependence and extensive colitis showed a significant association with UC without concomitant colonic diverticulosis (Table 3).

## 4. Discussion

This study was undertaken to evaluate whether UC patients with concomitant colonic diverticulosis differ from those without this condition in terms of the disease’s course and characteristics. Our data indicate that concomitant diverticulosis marks a subgroup of UC patients with a distinct clinical course. Indeed, the number of UC patients with more than one relapse per year and the percentages of patients with steroid-dependence and requiring biologics and immunosuppressants were lower in the group of UC patients with concomitant diverticulosis (cases) as compared to those seen in UC patients without diverticulosis (controls). On multivariate analysis, steroid dependence was still more frequent in UC patients without concomitant colonic diverticulosis. It seems thus fair to conclude that co-existence of UC and diverticulosis identifies patients with a less aggressive clinical UC course. Factors accounting for such findings remain to be ascertained even though it is likely that differences in terms of clinical course between cases and controls can, at least in part, rely on the different extension of the lesions in the two groups. Indeed, UC patients with concomitant colonic diverticulosis were less likely to have an extensive colitis, which is known to associate with a more severe course of the disease. In recent years, new insights have emerged regarding the qualitative and the quantitative changes in microbiota composition and diversity, which appear to be crucial for the occurrence and the persistence of symptoms in patients with colonic diverticulosis [11]. Consistently, the poorly absorbable drug rifaximin, which favours the growth of bacteria beneficial to the host without altering its overall composition, has been used with success in the prevention of exacerbations of colonic diverticulosis [12,13]. In addition, rifaximin is able to down-regulate the inflammatory response triggered by the gut microbes through inhibition of the activation of the transcription factor nuclear factor-κB and of the induction of inflammatory cytokines, such as Tumor Necrosis Factor and interleukin-1β [14,15]. Since all our UC patients with colonic diverticulosis were receiving cyclically (7 days per month at the dose of 400 mg twice per day) rifaximin, it is conceivable that such a treatment can have contributed to limit the inflammatory response in this subgroup of patients. To be tested, this hypothesis would require comparison of such patients with a cohort of UC patients with concomitant colonic diverticulosis taking no rifaximin. However, in our centre, all the patients with colonic diverticulosis are regularly treated with rifaximin. To date, only two studies have analysed the characteristics of UC in patients with concomitant diverticulosis even though both studies were aimed to primarily assess the prevalence of diverticula in UC. Nascimbeni and co-workers showed that diverticular disease development in UC patients was associated with a late onset of the inflammatory bowel disease [9]. Moreover, the authors showed that UC patients with diverticula had a significantly lower frequency of flares per year. Among these patients, there was a prevalence of left-sided colitis over proctitis and pancolitis, whereas, among those patients without diverticula, there was a trend toward a higher rate of extensive colitis. In contrast, Kinnucan and co-workers found no differences in terms of disease extent between UC patients with concomitant diverticulosis and those without diverticulosis [7]. Our study differs from these two reports as the main objective of the present study was to compare the UC characteristics and clinical course of the two populations of UC patients, who were homogenous for age, thus limiting a potential bias that could influence not only the development of colonic diverticulosis but also some UC features [16]. 

We are aware that our study has some limitations. This was a retrospective study in which identification of the cases and the controls was based on previous records, and the follow-up time occurred completely in the past before the design of the study. Therefore, we cannot exclude the possibility that there was some minimal degree of imprecision on the definition of some UC characteristics (e.g., severity and duration of the flare-ups). Moreover, exacerbation of UC was defined exclusively using clinical parameters given the lack of information about inflammatory markers (e.g., faecal calprotectin) and of endoscopic/histologic data at any scheduled visit of the patients. Finally, the duration of follow-up, though not different between the two groups (5 and 7 years, respectively), was relatively short, and this could have influenced some outcomes. A potential confounder in the diagnosis of UC and concomitant diverticulosis is the diagnosis of segmental colitis associated with diverticulitis (SCAD), an inflammatory condition affecting the colon in segments, which are also affected by diverticulosis (namely, the sigmoid colon) [3]. In our study, to ensure that patients with limited UC did not in fact have a SCAD, all UC diagnoses were confirmed through a histologic analysis of rectal biopsy samples, which showed immune-morphologic changes consistent with UC. Moreover, all UC diagnoses were made in patients with lesions involving the rectum, which is a major criterium to exclude the presence of SCAD. Although this study was not designed to assess the frequency of diverticula and to evaluate the natural history of colonic diverticulosis in UC, the characteristics of colonic diverticulosis in our group of UC patients were similar to those of the general population, with a comparable number of episodes of acute diverticulitis and of surgery for complicated diverticular disease. Our findings could help stratify UC patients and predict disease course, at least in a subgroup of patients. This could provide an additional tool to identify which patients would more likely benefit from major disease modifying treatments. 

In conclusion, UC patients with concomitant colonic diverticulosis are less likely to have an extensive disease and to be steroid-dependent. 

## Figures and Tables

**Table 1 jcm-10-01192-t001:** Characteristics of diverticula in patients with ulcerative colitis (UC) and concomitant diverticular disease.

CHARACTERISTICS	UC with Diverticulosis
Number of patients	92
Timing of diagnosis of diverticular disease compared to that of UC *n* (%)	
Previous	14 (15.2)
Subsequent	57 (61.9)
Concomitant	21 (22.8)
Diverticula localization *n* (%)	
Left colon	70 (76.1)
Whole colon	22 (23.9)
History of acute diverticulitis *n* (%)	11 (11.6)
Surgery for diverticular disease *n* (%)	4 (4.2)

**Table 2 jcm-10-01192-t002:** Demographic and clinical characteristics of UC patients with diverticulosis and UC patients without diverticulosis upon propensity score matching. Numbers indicate the number of patients for each variable.

CHARACTERISTICS	UC with Diverticulosis	UC without Diverticulosis	*p*-Value
Number of patients	92	153	
Gender female *n* (%)	37 (40.2)	46 (30.0)	0.2
Median Age (range) years	71 (48-92)	67 (48-84)	0.89
Smoking habit *n* (%)			
Current smokers	10 (10.8)	11 (7.1)	0.11
Former smokers	32 (34.7)	46 (30.06)	
Appendectomy *n* (%)	10 (10.8)	6 (3.9)	0.056
Mean disease’s duration (years)	19	16	0.008
UC extent *n* (%)			
E1	19 (20.6)	14 (9.1)	<0.001
E2	48 (52.1)	69 (45.0)	0.068
E3	25 (27.1)	70 (45.7)	<0.001
Steroid dependence *n* (%)	8 (8.6)	48 (31.3)	<0.001
Steroid refractoriness *n* (%)	1 (1.0)	7 (4.5)	0.09
Need of ISS *n* (%)	9 (9.7)	37 (24.1)	0.007
Need of biologics *n* (%)	3 (3.2)	26 (16.9)	<0.001
>1 relapse/year *n* (%)	6 (6.5)	16 (10.4)	0.013
Hospitalization *n* (%)			
0	64 (69.5)	113 (74.3)	0.44
≥1	28 (30.4)	40 (26.1)	
History of colorectal surgery *n* (%)	13 (14.1)	13 (8.4)	0.068
EIM *n* (%)	12 (13.0)	32 (20.9)	0.17
Years of follow-up median (range)	7 (1–19)	5 (1–19)	0.9

Abbreviations: UC: ulcerative colitis; ISS: immunosuppressants; EIM: extra-intestinal manifestations.

**Table 3 jcm-10-01192-t003:** Multivariate analysis of factors differently associated with ulcerative colitis (UC) patients with or without concomitant colonic diverticulosis.

VARIABLES	UC with Diverticulosis	UC without Diverticulosis	*p*-Value
Number of patients	92	153	
UC extent E3 *n* (%)	25 (27.1)	70 (45.7)	<0.001
Steroid dependence *n* (%)	8 (8.6)	48 (31.3)	0.028
Need of ISS *n* (%)	9 (9.7)	37 (24.1)	0.51
Need of biologics *n* (%)	3 (3.2)	26 (16.9)	0.31

## Data Availability

Data are available from the corresponding author upon reasonable request.

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
