# Peer review of "Natural History of Ulcerative Colitis with Coexistent Colonic Diverticulosis"

_jcm, 2021, doi:10.3390/jcm10061192_

Round 1

Reviewer 1 Report

The authors should comment on the significance of their findings. How would this data be applicable to the management or UC patients with diverticulosis? What kind of surgery was performed for diverticular disease in the 4 patients? Why was a restorative proctocolectomy not considered and did these patients have any flare up of their UC after surgery?

Author Response

We would like to thank the reviewer for her/his evaluation and helpful suggestions. In response to the specific points raised by this reviewer:

The authors should comment on the significance of their findings. How would this data be applicable to the management or UC patients with diverticulosis? What kind of surgery was performed for diverticular disease in the 4 patients? Why was a restorative proctocolectomy not considered and did these patients have any flare up of their UC after surgery?

Response: We have added in the conclusions section the possible implications of our findings in the management of UC patients. We have also specified in the results section what kind of colonic surgery was performed in patients with complicated colonic diverticulosis and highlighted that, for these patients, the number of UC flares were considered globally, before and after surgery, depending on patients’ history and time of diagnosis.

Reviewer 2 Report

Marafini and Coworkers present an interesting observational retrospective study assessing the phenotype of UC in patients with/and without diverticula

Propensity matched analysis for age was conducted. Patients with right-sided diverticulosis were excluded, as these have some other characteristics, this seems reasonable (and they are much more common in asian countries).

Overall, Patients with UC and diverticulosis presented with a less severe course of UC.

Overall the manuscript is written well and the topic of interest.

Several  major issues should be addressed before the manuscript could be recommended for publication:

In this cohort, all patients used cyclic rifaximin, the authors should add more comments on the rationale for this. What dose was used?

There is major doubt, that the observed changes are due to the rifaximin use.

It therefore seems necessary to provide another cohort of patients with diverticulosis and no rifaximin-use.

SCAD and UC can be hard to distinguish, this limitation needs to be explained in further depth. Is a number to patients with SCAD available for comparison of the characteristics?

Author Response

We would like to thank the reviewer for her/his evaluation and helpful comments and suggestions. In response to the specific points raised by this reviewer:

In this cohort, all patients used cyclic rifaximin, the authors should add more comments on the rationale for this. What dose was used? There is major doubt, that the observed changes are due to the rifaximin use. It therefore seems necessary to provide another cohort of patients with diverticulosis and no rifaximin-use.

Response: We have specified the dose of rifaximin currently used and we have provided the rationale and references for the use of this low-absorbable antibiotics in patients with colonic diverticulosis. Since all our patients are on rifaximin  it is not possible to provide data relative to a cohort of patients with diverticulosis receiving no rifaximin. However, we agree with the reviewer’s comment that  difference between the two groups could be due to rifaximin and, indeed, this point was commented.

SCAD and UC can be hard to distinguish, this limitation needs to be explained in further depth. Is a number to patients with SCAD available for comparison of the characteristics?

Response: We have taken into account the precious suggestion raised by the reviewer and have highlighted in the text how we have excluded patients with SCAD. All our UC patients had lesions involving the rectum as well, which is a major criterium to exclude a diagnosis of SCAD. This observation was added in the conclusions section of the revised version of the manuscript.  

Round 2

Reviewer 1 Report

All of the UC patients with colonic diverticulosis were receiving rifaximin. This may attribute to less extensive disease and steroid dependence. This is mentioned in the discussion but should be emphasized in the limitations of the study

Author Response

All of the UC patients with colonic diverticulosis were receiving rifaximin. This may attribute to less extensive disease and steroid dependence. This is mentioned in the discussion but should be emphasized in the limitations of the study.

Response: We would like to thank the reviewer for the suggestion. We have discussed this point among the limits of the study.

Reviewer 2 Report

manuscript was improved and can now be recommended for publication

more comments on the rüfaximin-therapy and rationale should be added before though

Author Response

Manuscript was improved and can now be recommended for publication. More comments on the rüfaximin-therapy and rationale should be added before though.

Response: We would like to thank the reviewer for the positive evaluation. The rationale for using rifaximin in patients with colonic diverticulosis has been indicated in the discussion section and it is underlined in the revised version of the manuscript.